# Quality of Life Following Vancouver Type B Periprosthetic Femoral Fractures: A Cross-Sectional Study

**DOI:** 10.3390/medicina61122159

**Published:** 2025-12-04

**Authors:** Nemanja Gvozdenović, Miodrag Vranješ, Igor Lekić, Sveto Bjelan, Andrijana Ćorić

**Affiliations:** 1Faculty of Medicine, University of Novi Sad, Hajduk Veljkova 3, 21137 Novi Sad, Serbia; nemanja.gvozdenovic@mf.uns.ac.rs (N.G.); miodrag.vranjes@mf.uns.ac.rs (M.V.); 911077d24@mf.uns.ac.rs (I.L.); sveto.bjelan@mf.uns.ac.rs (S.B.); 2Clinic for Orthopedic Surgery and Traumatology, University Clinical Center of Vojvodina, Hajduk Veljkova 1, 21137 Novi Sad, Serbia; 3Clinic for Plastic and Reconstructive Surgery, University Clinical Center of Vojvodina, Hajduk Veljkova 1, 21137 Novi Sad, Serbia

**Keywords:** quality of life, periprosthetic femoral fracture, total hip replacement, Vancouver classification

## Abstract

*Background and Objectives*: Total hip arthroplasty (THA) is a common orthopedic procedure. It helps restore mobility and reduce pain in patients with hip joint disorders. Periprosthetic femoral fracture (PFF) is an acute complication that may occur after primary THA. The rate of PFFs after primary total hip replacement is approximately 1%. The aim of this study was to assess the overall quality of life of patients following PFF surgery. *Materials and Methods*: This cross-sectional study included 60 patients with PFFs of Vancouver type B (32 females and 28 males, respectively), with a mean age of 73.02 ± 8.97 years and 30 controls who underwent primary THA. Quality of life was assessed at least 12 months postoperatively using the validated Serbian SF-36 questionnaire and clinical examination. *Results*: Older age correlated with declines in Physical and Emotional functioning, Vitality (Energy/fatigue), and Social activities (overall SF-36: r = −0.619, *p* < 0.01). Patients who underwent femoral stem revision with osteosynthesis (B2 and B3) showed better quality of life compared to those who underwent osteosynthesis alone (B1) in General health perceptions (t = −2.266, *p* = 0.027) and Physical functioning (t = 2.526, *p* = 0.014). Patients after PFF surgery had lower postoperative quality of life compared to those who underwent primary THA (overall SF-36: 66.68 ± 15.60 vs. 84.10 ± 14.65, t = −5.092, *p* < 0.0005). *Conclusions*: Patients with PFF have a lower quality of life than those after primary THA, while combined stem revision and osteosynthesis yield better outcomes than osteosynthesis alone.

## 1. Introduction

Total hip arthroplasty (THA) is one of the most common procedures in orthopedic surgery, which helps restore mobility and reduce pain in patients with hip joint disorders [1]. The demand for THA is expected to increase by 43% to 70% by 2030 [2].

Periprosthetic femoral fracture (PFF) is an acute complication that may occur after primary THA [3]. The rate is approximately 1% [4], and is estimated to increase by 4.6% every 10 years over the next 30 years [5].

As the number of THA procedures increases, so does the incidence of complications. PFF accounts for 3.5–5.0% of all THA-related complications [6]. The most frequent causes are aseptic implant loosening, instability, and infection [7]. PFF can occur intraoperatively or postoperatively (1% vs. 5%) and is more common with uncemented prostheses (cemented 0.3%, uncemented 5.4%) [6]. Fractures typically occur in elderly patients with pre-existing health issues, often after a ground-level fall. Risk factors for PFF include osteoporosis, age ≥ 64 years, obesity, prior femoral fracture, rheumatoid arthritis, aseptic stem loosening, heart disease, peptic ulcers, and multiple previous hip surgeries [8,9,10].

The management of PFFs requires an experienced trauma and orthopedic surgeon due to its complexity and high risk of postoperative complications. These complex surgeries are often accompanied by a high complication rate (failure rate) and a high mortality rate (9% in the first year and 60% after 5 years) [8]. Patients diagnosed with PFFs are at risk of developing infection, pressure ulcers, fracture nonunion, prosthesis loosening, thromboembolic complications, depression, and other psychological disorders [8,9,10,11,12]. PFFs are classified according to the Vancouver Classification System, which stratifies them based on location, implant stability and bone quality, thus providing the treatment recommendations according to the specific fracture type (Table 1) [13,14,15].

Most PFFs are type B (B1: 29%, B2: 53%, B3: 4%), as they are extremely challenging to treat [16]. The recommended treatment approach for all type B fractures is surgery; type B1 requires open reduction and internal fixation (ORIF), while types B2 and B3 necessitate a revision prosthesis in conjunction with ORIF. However, numerous factors guide treatment in everyday clinical practice, including the patient’s condition with comorbidities, functional status, the surgeon’s experience, bone quality, and the type of fracture.

PFFs often lead to a decreased quality of life [17]. The most common outcome measures for PFFs are mortality, healing rate, wound healing time, and surgery-specific outcomes (blood loss, operative time, and adverse events) [18]. A large number of studies have analyzed functional outcomes and quality of life after PFF [5,17,19,20,21,22,23] yet there is still a gap in understanding the quality of life of patients with these fractures [18,24,25]. Most of them have a retrospective design, with limited sample sizes. Qualitative studies are poorly represented [21], and most research focuses predominantly on orthopedic scores and quantitative indicators of functionality. Therefore, this study aimed not only to examine the orthopedic recovery of patients with Vancouver type B PFF, but also to evaluate their quality of life, encompassing their subjective experience of the recovery process, including fear, duration of recovery, the impact of prolonged stays in orthopedic and rehabilitation clinics, in order to assess whether and to what extent patients return to their pre-fracture daily activities.

### The 36-Item Short Form Survey (SF-36)

The 36–Item Short Form Survey (SF-36) is a widely used instrument designed to measure health outcomes. Originally developed as part of the Medical Outcomes Study, its primary purpose is to provide an objective assessment of an individual’s quality of life [26]. The SF-36 consists of thirty-six questions. Thirty-five are grouped into eight domains: Physical activity limitations due to health problems; Social activity limitations due to physical or emotional problems; Role activity limitations due to physical health problems; Bodily pain; General mental health; Role activity limitations due to emotional problems; Vitality (energy and fatigue); and General health perceptions [26].

The Physical Component Summary (PCS) includes Physical functioning, Role limitations due to physical health problems, Bodily pain, and General health perceptions. The Mental Component Summary (MCS) includes Vitality, Social functioning, Role limitations due to personal or emotional problems, and Emotional well-being. One question is intended to compare general quality of life now compared to a year ago.

Through this comprehensive approach, the SF-36 provides a detailed understanding of an individual’s overall health and well-being.

The results of each domain (1–5 points) are transformed into standardized values 0–100. Scoring is calculated for each domain, where a higher score indicates a better quality of life. For each domain, a score of 0–33 indicates poor quality of life, 33–66 is considered good quality of life, and 66–100 as excellent quality of life [27].

The reliability and validity of the SF-36 have been confirmed in numerous studies, making it one of the most reliable and most frequently used research instruments for assessing the quality of life of patients with various diseases [28,29,30].

## 2. Materials and Methods

### 2.1. Study Design and Eligibility Criteria

This was a cross-sectional observational study conducted at the Clinic for Orthopedic Surgery and Traumatology, University Clinical Center of Vojvodina.

An increase in the number of PFFs has also been observed at our Clinic over the last years (Figure 1).

Inclusion criteria (experimental group—PFF):Age ≥ 65 yearsVancouver type B proximal PFFSurgical treatment at our clinic between January 2016 and December 2022Voluntary consent to participate

Exclusion criteria (experimental group—PFF):Age < 65 yearsDeath prior to follow-upIntraoperative fracturesCognitive impairment preventing independent survey completion

A total of 63 consecutive patients with Vancouver type B periprosthetic femoral fractures were initially screened for eligibility. Three patients were excluded due to death prior to follow-up or cognitive impairment (Figure 2).

A final sample included 60 patients with PFFs of Vancouver type B (32 females and 28 males, 53% and 47%, respectively), with a mean age of 73.02 ± 8.97 years) who underwent surgery between January 2016 and December 2022 at our center.

The Vancouver classification was used to assess fracture types [14,15]. B1 fractures were surgically treated with ORIF using locking compression plates (LCP). B2 and B3 fractures were treated with revision prosthesis combined with ORIF. The sample included 37 patients with B1, 19 with B2, and 4 with B3-type fractures.

The control group consisted of 30 consecutive patients over 65 years from the same clinic who underwent primary THA for traumatic femoral neck fractures between 2016 and 2022 (17 females, 13 males; mean age 71.36 ± 8.53 years). Patients were matched to the PFF group by age, sex, ASA Physical Status Classification System [31], and trauma status to ensure sample homogeneity.

Baseline demographic and clinical characteristics are summarized in Table 2.

### 2.2. Quality of Life Assessment and Data Collection

Quality of life was measured using the validated Serbian translation of 36-Item Short Form Survey Instrument (SF-36) [26,32] with a clinical examination at least 12 months after surgery. The study was conducted by surveying patients during routine follow-up visits at the specialist polyclinic of the University Clinical Center of Vojvodina. Patients completed the questionnaire in a private room. A registered nurse was available to assist if needed.

Patient data were collected from medical records (hospital files, the Clinical Information System—internal hospital system, and the Picture Archiving and Communication System—PACS).

### 2.3. Statistical Data Analysis

Statistical data processing and analysis were performed using SPSS Statistics version 25.0 (IBM Corporation, Armonk, NY, USA). Descriptive statistics (means and standard deviations) were calculated for all variables. All SF-36 domain raw scores (1–5) were transformed to standardized 0–100 scores according to the scoring manual.

Pearson correlation was used to determine the correlation between SF-36 domains. The normality of data distribution was checked with the Kolmogorov–Smirnov test, and independent samples *t*–tests were conducted to compare differences in quality of life between subgroups according to sex, operated leg, and PFF type (B1 vs. B2/B3), as well as between the experimental group (periprosthetic fracture) and the control group (primary THA). Statistical significance was set at *p* < 0.05.

## 3. Results

### 3.1. Age and Quality of Life

Pearson’s correlation analysis (Table 3) showed significant associations between age and most quality of life domains.

Age correlated positively with General health perceptions (r = 0.768) and Bodily pain (r = 0.735). Negative correlations were observed between age and Physical functioning (r = −0.806), Role limitations due to physical health problems (r = −0.731), Role limitations due to personal or emotional problems (r = −0.687), Social functioning (r = −0.687), Vitality (Energy/fatigue) (r = −0.762), Emotional well-being (r = −0.827), and the Overall SF-36 score (r = −0.619). These findings indicate that older patients reported better subjective perceptions of general health and less bodily pain, with a decline in physical and emotional functioning, energy and social activities.

### 3.2. Sex Differences

Independent *t*–test results (Table 4) showed no significant differences in health status or treatment outcomes between males and females after PFF surgery. Mean SF-36 values were comparable across all domains, indicating that sex did not significantly affect postoperative quality of life in this sample.

### 3.3. Differences in Quality of Life According to Operated Leg

No significant differences were observed between patients with left- and right-sided fractures (Table 5). 

### 3.4. Quality of Life and the Type of PFF Revision

Patients who underwent femoral stem revision combined with osteosynthesis (B2 and B3 fractures) reported higher values in several quality of life domains compared to those who underwent osteosynthesis alone (B1 fractures) (Table 6).

### 3.5. Comparison with Primary THA

Patients after PFF surgery had significantly lower overall SF-36 scores compared to those who underwent primary THA (*t* = −5.092, *p* < 0.0005). The largest difference was observed in the General Health Perceptions domain (*t* = 4.717, *p* < 0.0005) (Figure 3, Table 7).

## 4. Discussion

This study demonstrates that the overall quality of life in patients following surgery for PFFs remained significantly lower than that of patients who underwent primary THA. Domains with the lowest scores were physical role limitations and pain. This highlights the enduring effects of complex trauma and revision surgery on physical functioning. Patients who underwent osteosynthesis combined with femoral stem revision (B2 and B3 fractures) reported better outcomes in terms of general health status, energy levels, and pain management than those who underwent osteosynthesis alone (B1 fractures). This suggests that appropriately selected and executed revision procedures may lead to improved functional recovery and enhanced self-perceived health status.

In our study, a total of 60 patients with PFF were included, along with 30 patients who underwent primary THA (control group). The average age of the patients in experimental group was 73 years and 71 years in the control group, which is significant enough to impact their overall quality of life. Some correlations in our study, such as older age being associated with better perceptions of general health and lower pain levels, may seem counterintuitive. This may be a consequence of changes in health expectations, the need for consistency in self-perception and the pursuit of positive self-enhancement, which are common in elderly adults and may evolve over time after treatment [33,34]. These findings can be explained by the concept of ‘response shift’, which is defined as a change in one’s self-evaluation resulting from changes in internal standards, values, or the conceptualization of the target construct [35].

The severity of the injury, the treatment method, and any comorbid conditions influence both treatment outcomes and the quality of life [36,37].

Recommended surgical treatments for PFFs include replacing the femoral shaft and using a longer implant to reconnect the fractured segments. ORIF alone has generally been contraindicated for B2 fractures due to the risk of non-union fracture associated with a loose stem, which often requires long-term immobilization and may lead to the need for further revisions. However, if ORIF could facilitate successful fracture healing without necessitating revision, it would significantly enhance treatment outcomes. Reducing operation time and simplifying the procedure may particularly benefit younger patients who are more likely to require subsequent revisions. Furthermore, ORIF could lower implant costs and decrease the time spent in the operating room [38].

In type B2 PFFs, particularly around cemented polished stems, implant stability can be significantly compromised even with minimal damage to the cement mantle. This occurs because the stabilizing force relies on the integrity of the cement mantle, which is responsible for transferring the load to the cortical walls. Consequently, this significantly impacts treatment outcomes, slows recovery, and diminishes the patient’s overall quality of life [39].

Timely surgery has been shown to significantly improve the prognosis of elderly patients with femoral fractures. It may contribute to reduced perioperative complications, better postoperative recovery, and improved quality of life [40].

In a study of 221 patients with PFFs, Pavlou et al. [41] reported that these fracture patterns require revision with a long stem prosthesis. The stem should span the fracture level by at least two cortical diameters, which is a general rule in surgical practice.

Pavlou et al. [41] and Gitajn et al. [42] report on the benefits of revision surgery for Vancouver B fractures, with better functional outcomes and lower mortality. In contrast, Antoniadis et al. [43] and Joestl et al. [44] reported more favorable outcomes in patients who underwent osteosynthesis alone, with sufficient fragment stability and without the need for prosthesis replacement. Solomon et al. [38] point out that osteosynthesis can achieve similar results to revision surgery in carefully selected patients with a stable femoral stem.

These different findings underscore the importance of an individualized approach: the assessment of prosthesis stability, bone quality and the general condition of the patient remains crucial for the treatment decision. The literature suggests that the choice of fixation technique and stem type significantly influence the outcome. For example, TFMT (Tapered Fluted Modular Titanium) stems have better biomechanical stability compared to standard cemented CB prostheses, which reduces the risk of reoperation and allows a faster return to physical activity. In line with these findings, the results indicate a statistically significant improvement in quality of life in patients undergoing stem replacement compared to those who underwent osteosynthesis alone, which has important clinical implications [39].

A similar age distribution of the patients was also found in the study by Olivo-Rodríguez et al. [11] where the mean age for PFF was 74, but with a significantly smaller number of subjects, 15. These findings are in line with the predictions from the study by Poulsen et al. [45] who indicated an expected increase in the number of patients undergoing primary THA, and therefore an increase in the incidence of PFFs. Thien et al. [46] suggest that the increased incidence of this pathology may be partly attributed to older patients with poorer bone quality, as well as younger patients with greater physical activity demands.

The application of newer surgical techniques, such as stem replacement and osteosynthesis of the femur, leads to better outcomes of surgical treatment and a better quality of life [47,48,49,50].

Patients with osteosynthesis without femoral stem revision have a poorer quality of life than those who undergo osteosynthesis with femoral stem revision. Khan et al. [51] performed a systematic review of 22 studies including 510 fractures classified as B2 and B3 and found that 12.6% of B2 and 4.8% of B3 fractures underwent IF alone. These were associated with a higher reoperation rate compared with revision arthroplasty. This highlights the importance of assessing stem stability during preoperative workup and intraoperative stem testing and ensuring that revision implants are available during surgery.

Ricci et al. reported that the results after surgery for distal PFFs are poorer and associated with a high mortality rate [52].

A large registry-based epidemiological study of over 3 million patients found that revision total hip arthroplasty (rTHA) had a significantly higher complication rate than primary THA (39.46% vs. 27.32%). Postoperative anemia was the most common complication in both groups. The most common indications for rTHA were also dislocation/instability (21.85%) and mechanical loosening (19.74%). The authors point out that the increased risk of complications in revision surgery is partly due to the higher number of comorbidities (e.g., high blood pressure, chronic lung disease) and the technically more complex nature of the procedure itself. Therefore, rTHA is a much more complex procedure with higher risks for the patient, which is to some extent comparable to results of this study. In patients after PFFS (which usually require a revision prosthesis), the results indicate lower scores in several domains of quality of life compared to the control group after primary THA [2].

We further analyzed whether gender, age, and the operated side affect the quality of life. Considering the operated side, in our study the incidence of right hip disease is 56.6%. On the other hand, left-sided fractures were less frequent (43.4%) compared to right-sided (56.6%), and based on the results obtained in relation to the operated leg, there are no significant differences in the quality of life. No statistically significant differences were observed in the total SF-36 score between males and females, (males 68.64/100, females 64.96 ± 15.91, respectively).

Although we did not use the EQ-5D-3L scale, we can compare our results with those of Nieboer et al. [18], who reported that male patients had slightly better postoperative outcomes than women, whereas in our case, there were no such differences. Pavlović et al. [25] showed that patients with revision arthroplasty for Vancouver B2/B3 fractures usually regain mobility and a higher quality of life, with an EQ-5D-5L index of around 0.8, which is consistent with our finding that patients with B2/B3 fractures and combined osteosynthesis have better SF-36 scores than patients with B1 fractures and osteosynthesis alone.

In a study by Zampelis et al. [53], which was conducted on 45 patients (27 males and 18 females), the results showed that men had better functional outcomes than women. As a possible explanation for this result, they stated that men and women value their health differently, as well as that their postoperative expectations are different. Compared with the findings of study by Märdian et al. [54], who analyzed the quality of life after periprosthetic knee fractures, significant differences are observed. In their study, the average SF-36 score was 41 ± 6, indicating a more pronounced impaired quality of life compared to our study group. Also, only 20% of the patients from that study were able to move independently, while the rest required some kind of assistance, which further confirms a greater functional deficit after a knee fracture. Their WOMAC scores also indicate high levels of pain, stiffness, and limitations in daily activities. In contrast to our study, they did not find a significant influence of the type of fracture or comorbidity on the final outcome, while we clearly identified a difference in outcomes depending on the method of treatment and type of fracture. This difference can be partially explained by the larger number of subjects in our study (total of 90 patients vs. 25), but both studies confirm that PFFs significantly impair quality of life, especially in the domain of physical functioning.

Although we did not directly examine risk factors for PFFs, findings from a retrospective cohort study by Ding et al. [8] suggest that some key factors, such as older age, osteoporosis, and previous femoral neck fractures, are associated with a higher incidence of PFF. These risk factors are important because they often occur in patients with reduced functional capacity and therefore may also affect quality of life after treatment. The reduced quality of life in patients with PFF may be partially explained by the presence of these risk factors and comorbidities, which further complicate recovery and affect perceptions of health and function.

It is important to emphasize that the choice of the appropriate surgical method can significantly affect functional recovery and therefore the quality of life of patients, which is consistent with our findings [55].

The results of this study are comparable with the findings of a study that analyzed the impact of fracture-related infections (FRI). That study showed that FRI leads to a significant deterioration in several domains of quality of life, including pain, mobility, and daily activities, as well as an increased rate of complications [56].

In addition, our results indicate that patients who underwent a combination of femoral stem replacement and osteosynthesis have significantly better outcomes in the domains of General health perceptions (*p* = 0.027), Physical limitations (*p* = 0.014), Bodily pain (*p* = 0.042), and Energy/fatigue (*p* = 0.009), compared to those who underwent osteosynthesis alone. These findings are consistent with the results of a study comparing different treatment methods for B2 fractures, where long-term stabilization (LSRIF) resulted in better outcomes in Harris Hip Score and SF-36 physical and mental scores compared to ORIF [3]. This indicates that the choice of surgical technique must be tailored to the individual patient’s condition, not just the fracture morphology.

The study by Cursaru et al. [57] highlights the importance of a multidisciplinary and personalized approach to the treatment of PFF, indicating the potential of cortical allografts, ORIF, or revision depending on bone quality and implant stability. These observations are consistent with our view that the therapeutic algorithm must be flexible and based on a comprehensive assessment.

Finally, our findings highlight the necessity for early, individualized surgical planning and comprehensive rehabilitation for patients with PFFs, particularly considering the unique characteristics of this population, which often includes elderly individuals. It is essential for healthcare professionals to address both the physical and psychological aspects of postoperative recovery. Standardized, valid, and reliable research instruments specifically designed to assess quality of life, such as the SF-36, can offer valuable insights into patient-centered outcomes.

### 4.1. Clinical Implications

Early surgery (within 48 h of hospital admission) is associated with significant benefits in terms of lower rates of postoperative complications and risk of death and may provide better functional outcomes [58].

In addition, an individualized treatment approach that takes into account risk factors such as osteoporosis and general health status can significantly improve the rehabilitation process. An adequate therapeutic approach to patients with osteoporosis can reduce the risk of PFF after THA, which emphasizes the need for early diagnosis and intervention in patients at increased risk [59].

A comprehensive multidisciplinary approach has recently become essential for rehabilitation programs. Interdisciplinary teams optimize patients’ medical, psychosocial, and physical capacities, including plans for early discharge. Since most femoral fractures occur in older patients, a multidisciplinary rehabilitation strategy that incorporates various specialists is crucial. To ensure the best outcomes, this treatment should commence as soon as the patient is hospitalized and continue throughout the rehabilitation process after discharge [60].

There is a great need to standardize the surgical approach and integrate multidisciplinary protocols to improve functional treatment outcomes and quality of life in patients with PFFs.

### 4.2. Study Limitations

The relatively small number of patients in both groups may restrict the statistical power and generalizability of the findings to the broader population. The study population was limited to patients who survived and were able to complete the SF-36 survey. Therefore, the results may be influenced by survival and selection bias. Although multivariate analysis was not performed, unadjusted comparisons between groups may be influenced by other factors. Differences in outcomes between B1 and B2/B3 fracture types may be due in part to patient selection, rather than solely to the type of treatment. B2/B3 fractures usually occur in patients with a more complex clinical profile, the presence of comorbidities, poorer bone quality, as well as poorer preoperative functional status, which may contribute to less favorable outcomes regardless of the type of surgery.

Additionally, some patients may have been biased when self-assessing their quality of life. They may have been less precise, anxious during the examination conducted by healthcare professionals, or provided socially desirable responses. In the context of evaluating quality of life, it is crucial to consider socioeconomic factors and the availability of rehabilitation services, as these can significantly influence treatment outcomes.

## 5. Conclusions

Patients following PFF demonstrate a lower quality of life compared to a control group of patients who have undergone THA after treatment, primarily due to multiple comorbidities and poorer health status. Additionally, patients who underwent both prosthesis replacement and femur osteosynthesis reported significantly better quality of life after treatment reported a better quality of life compared to those who underwent femur osteosynthesis alone. The increase in the incidence of PFFs due to an aging global population requires a holistic approach, individualized surgical planning, and optimized postoperative rehabilitation, which are the basis for improving patients’ functional recovery and quality of life.

More extensive prospective studies are needed to investigate long-term quality of life trends and to optimize treatment protocols for this growing and very sensitive patient population.

## Figures and Tables

**Figure 1 medicina-61-02159-f001:**
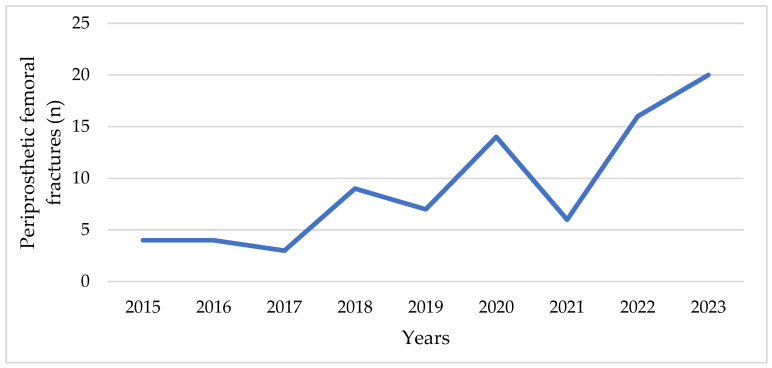
Number of periprosthetic fractures at the Clinic for Orthopedic Surgery and Traumatology of the University Clinical Center of Vojvodina in the period from 2015 to 2023.

**Figure 2 medicina-61-02159-f002:**
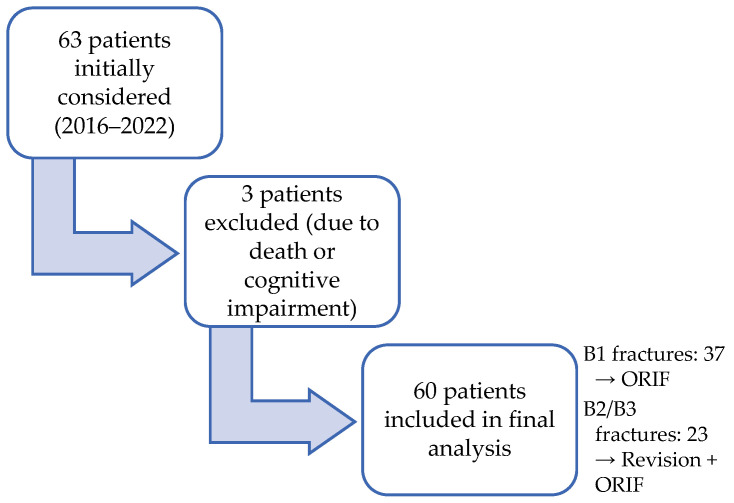
Patient recruitment flow (experimental group).

**Figure 3 medicina-61-02159-f003:**
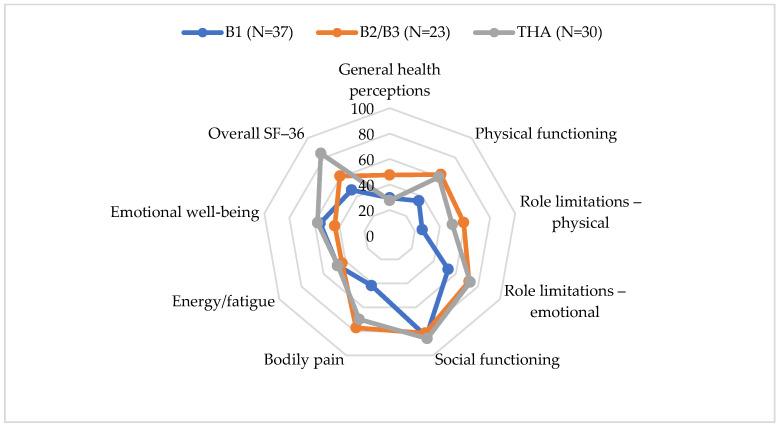
SF-36 scores by PFF type and THA.

**Table 1 medicina-61-02159-t001:** Modified Vancouver classification system for periprosthetic fractures.

Type	Description	Example
A	Involving apophysis, e.g., avulsion	Greater or lesser trochanter
B	Directly adjacent to implant	Femoral shaft fracture around stem
B1	Well-fixed implant	
B2	Loose implant and good bone stock	
B3	Loose implant and poor bone stock	
C	Distant to implant but within same bone	Distal femur fracture below stem
D	Interprosthetic—between two joint replacements at either end of long bone	Femoral shaft fracture between a hip and knee replacement
E	Involves two bones supporting one joint replacement	Combined acetabular and femur fracture around THR
F	At native joint surface which directly articulates with an implant	Acetabular fracture next to hip hemiarthroplasty

**Table 2 medicina-61-02159-t002:** Patient baseline characteristics and follow-up.

Parameter	PFF Group (*n* = 60)	THA Control (*n* = 30)
Sex (F/M) *n* (%)	32/28 (53.3/46.7)	17/13 (56.7/43.3)
Age (years), mean ± SD	73.02 ± 8.97	71.36 ± 8.53
Time to surgery (days), mean ± SD	3.2 ± 1.4	2.7 ± 1.2
Vancouver type (B1/B2/B3)	37/19/4	-
Surgical treatment	ORIF (B1), Revision + ORIF (B2/B3)	Primary THA
ASA status (II/III), *n* (%)	24/36 (40.0/60.0)	13/17 (43.3/56.7)
Follow-up (months), mean ± SD	25.1 ± 7.2	23.4 ± 5.1

**Table 3 medicina-61-02159-t003:** Correlation between patient age and quality of life after PFF surgery.

Domain	r	95% CI (Lower)	95% CI (Upper)
General health perceptions	0.768 **	0.646	0.851
Physical functioning	−0.806 **	−0.877	−0.702
Role limitations—physical	−0.731 **	−0.817	−0.607
Role limitations—emotional	−0.687 **	−0.782	−0.557
Social functioning	−0.687 **	−0.782	−0.557
Bodily pain	0.735 *	0.615	0.829
Energy/fatigue	−0.762 **	−0.842	−0.660
Emotional well-being	−0.827 **	−0.888	−0.738
Overall SF-36	−0.619 **	−0.730	−0.471

* Significance level 0.05, ** Significance level 0.01, r = Pearson’s correlation coefficient.

**Table 4 medicina-61-02159-t004:** Differences in quality of life of patients after PFF surgery by sex.

Domain	Male $\bar{x}$ ± SD	Female $\bar{x}$ ± SD	t	*p*	Cohen’s d	95% CI (d)
General health perceptions	42.0 ± 31.0	39.5 ± 32.0	−0.306	0.760	−0.079	−0.587 to 0.428
Physical functioning	54.0 ± 42.0	50.5 ± 43.0	0.317	0.752	0.082	−0.425 to 0.589
Role limitations—physical	46.0 ± 50.0	46.0 ± 50.0	−0.034	0.973	−0.009	−0.516 to 0.498
Role limitations—emotional	67.0 ± 44.0	63.0 ± 45.0	0.367	0.715	0.095	−0.412 to 0.602
Social functioning	83.25 ± 44.0	84.25 ± 42.0	0.367	0.715	0.095	−0.412 to 0.602
Bodily pain	72.6 ± 68.0	70.6 ± 70.0	−0.436	0.665	−0.113	−0.620 to 0.394
Energy/fatigue	45.0 ± 30.0	44.4 ± 26.0	−0.422	0.674	−0.110	−0.617 to 0.397
Emotional well-being	50.4 ± 32.0	50.8 ± 32.0	0.040	0.968	0.010	−0.497 to 0.517
Overall SF-36	68.64 ± 15.28	64.96 ± 15.91	0.909	0.367	0.236	−0.254 to 0.726

**Table 5 medicina-61-02159-t005:** Differences in quality of life of patients by operated leg.

Domain	Left $\bar{x}$ ± SD	Right $\bar{x}$ ± SD	t	*p*	Cohen’s d	95% CI (d)
General health perceptions	49.0 ± 36.0	52.5 ± 34.0	0.519	0.605	0.142	−0.400 to 0.685
Physical functioning	54.0 ± 42.0	60.0 ± 41.0	−0.686	0.494	−0.187	−0.728 to 0.354
Role limitations—physical	41.0 ± 49.0	54.0 ± 50.0	−1.254	0.213	−0.341	−0.882 to 0.201
Role limitations—emotional	61.0 ± 47.0	75.0 ± 41.0	−1.423	0.158	−0.386	−0.928 to 0.157
Social functioning	84.5 ± 48.0	81.5 ± 42.0	−1.426	0.157	−0.387	−0.929 to 0.156
Bodily pain	72.0 ± 61.0	72.6 ± 62.0	0.104	0.917	0.028	−0.513 to 0.569
Energy/fatigue	45.4 ± 23.0	44.8 ± 24.0	−0.213	0.832	−0.058	−0.599 to 0.484
Emotional well-being	50.8 ± 32.0	46.6 ± 34.0	−0.609	0.544	−0.165	−0.706 to 0.377
Overall SF-36	72.67 ± 17.49	72.29 ± 17.30	0.103	0.918	0.028	−0.513 to 0.569

**Table 6 medicina-61-02159-t006:** Differences by type of PFF revision.

Domain	B2 and B3 (Osteosynthesis in Conjunction with a Stem Replacement) (*n* = 23) $\bar{x}$ ± SD	B1 (Osteosynthesis) (*n* = 37) $\bar{x}$ ± SD	t	*p*	Cohen’s d	95% CI (d)
General health perceptions	47.5 ± 31.0	29.5 ± 28.5	−2.266	0.027 *	−0.501	−0.936 to −0.067
Physical functioning	62.5 ± 42.0	35.5 ± 37.5	2.526	0.014 *	0.599	0.128 to 1.070
Role limitations—physical	59.0 ± 49.0	26.0 ± 44.0	2.684	0.010 *	0.636	0.161 to 1.111
Role limitations—emotional	72.0 ± 42.0	53.0 ± 46.0	1.639	0.107	0.389	−0.101 to 0.879
Social functioning	81.5 ± 42.0	84.25 ± 46.0	1.639	0.107	0.389	−0.101 to 0.879
Bodily pain	77.0 ± 69.5	41.8 ± 62.0	−2.074	0.042 *	−0.486	−0.923 to −0.049
Energy/fatigue	43.2 ± 28.0	47.2 ± 23.0	2.839	0.006 *	0.672	0.199 to 1.146
Emotional well-being	43.8 ± 32.0	55.4 ± 30.0	2.797	0.007 *	0.662	0.191 to 1.132
Overall SF-36	60.8 ± 41.8	46.6 ± 39.8	1.22	0.227	0.287	−0.167 to 0.741

* Significance level 0.05.

**Table 7 medicina-61-02159-t007:** Differences in quality of life after PFF surgery (experimental group) and outcomes after THA (control group).

Domain	PFF $\bar{x}$ ± SD	THA $\bar{x}$ ± SD	t	*p*	Cohen’s d	95% CI (d)
General health perceptions	42.4 ± 31.3	27.5 ± 29.5	4.717	<0.0005 *	0.972	0.621 to 1.323
Physical functioning	54.0 ± 42.0	60.0 ± 37.5	−1.557	0.123	−0.331	−0.733 to 0.072
Role limitations—physical	46.0 ± 50.0	50.0 ± 50.0	−0.295	0.768	−0.063	−0.466 to 0.339
Role limitations—emotional	65.0 ± 45.0	73.0 ± 44.0	−0.772	0.442	−0.164	−0.566 to 0.239
Social functioning	83.0 ± 45.0	86.0 ± 43.0	−0.770	0.445	−0.163	−0.566 to 0.239
Bodily pain	72.5 ± 68.5	70.0 ± 43.0	0.875	0.384	0.185	−0.226 to 0.596
Energy/fatigue	45.0 ± 28.0	47.0 ± 8.0	1.874	0.065	0.345	−0.043 to 0.733
Emotional well-being	50.5 ± 32.0	57.5 ± 32.0	−1.630	0.107	−0.300	−0.701 to 0.101
Overall SF-36	66.68 ± 15.60	84.10 ± 14.65	−5.092	<0.0005 *	−1.031	−1.372 to −0.690

* Significance level 0.05.

## Data Availability

Dataset available on request from the authors.

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
