# Peer review of "Quality of Life Following Vancouver Type B Periprosthetic Femoral Fractures: A Cross-Sectional Study"

_medicina, 2025, doi:10.3390/medicina61122159_

Round 1
Reviewer 1 Report
Comments and Suggestions for Authors
- A summary table of baseline characteristics (sex, age, fracture type, follow-up period, etc.) would make the cohort easier to understand.
- A patient flow chart is also recommended to clarify case selection and dropouts (death, dementia), improving transparency.
- Table 1 includes Vancouver types D–F, which are not part of the original system. If you intend to keep them, please label it as “Modified Vancouver Classification”; otherwise, cite the original version only.
- In Table 2, older age shows higher General Health scores, which is counterintuitive—please verify the data or discuss this finding.
- In Table 5, General Health is higher in B1 (3.82) than in B2/B3 (3.09), yet the text states the revision group performed better; check for data or interpretation errors and revise accordingly.
- Similarly, Table 6 shows higher PFF scores than THA; please confirm.
- Clarify how the 30 primary THA patients were selected and whether any matching (e.g., PSM) was performed.
- SF-36 scaling is inconsistent: methods mention 0–100 normalization, but tables show 1–5 scales. Please standardize all domains to 0–100.
- Line 82: PPF appears to be a typo.
Author Response
Comments 1: A summary table of baseline characteristics (sex, age, fracture type, follow-up period, etc.) would make the cohort easier to understand.
Response 1: Thank you for pointing this out. We agree with this comment. Therefore, we have added a table summarizing baseline characteristics and follow-up information for both the PFF and THA control groups to clarify group selection and comparability of follow-up intervals. This change can be found in the revised manuscript on page 4-5, Materials and Methods section, Table 2.
Comments 2: A patient flow chart is also recommended to clarify case selection and dropouts (death, dementia), improving transparency.
Response 2: Thank you for pointing this out. We agree with this comment. Therefore, we have added a participant flow diagram in the Materials and Methods section, page 4, including total eligible fractures, exclusions (death, cognitive impairment), and the final sample.
Comments 3: Table 1 includes Vancouver types D–F, which are not part of the original system. If you intend to keep them, please label it as “Modified Vancouver Classification”; otherwise, cite the original version only.
Response 3: Thank you for pointing this out. We agree with this comment. Therefore, we have clearly labeled Table 1 as the ‘Modified Vancouver Classification System. This change can be found in the revised manuscript on page 2, Introduction section, line 61.
Comments 4: In Table 2, older age shows higher General Health scores, which is counterintuitive—please verify the data or discuss this finding.
Response 4: Thank you for pointing this out. We agree with this comment. Therefore, we have verified that the SF-36 scoring direction is correct, and higher scores indicate better health status. References have been added to support this and changes can be found in the revised manuscript on page 9, Discussion section, lines 231-238.
Comments 5: In Table 5, General Health is higher in B1 (3.82) than in B2/B3 (3.09), yet the text states the revision group performed better; check for data or interpretation errors and revise accordingly.
Response 5: Thank you for pointing this out. We agree with this comment. The data in Table 5 was checked and corrected. This change (Now Table 6) can be found in the revised manuscript on page 7, Results section, lines 200–201.
Comments 6: Similarly, Table 6 shows higher PFF scores than THA; please confirm.
Response 6: Thank you for pointing this out. We agree with this comment. We have verified the values (now Table 7). This change can be found in the revised manuscript on page 8, Results section, lines 211–213.
Comments 7: Clarify how the 30 primary THA patients were selected and whether any matching (e.g., PSM) was performed.
Response 7: Thank you for pointing this out. We agree with this comment. Therefore, we have clarified in the Materials and Methods section that the 30 primary THA patients were selected from those over 65 years undergoing surgery for degenerative hip disease and were matched to the PFF group by age, sex, ASA status, and trauma status. No additional matching or propensity score methods were applied. Changes can be found in the revised manuscript on page 5, Materials and Methods section, lines 132–135.
Comments 8: SF-36 scaling is inconsistent: methods mention 0–100 normalization, but tables show 1–5 scales. Please standardize all domains to 0–100.
Response 8: Thank you for pointing this out. We agree with this comment. Therefore, all SF-36 domains have been standardized to a 0–100 scale throughout the manuscript. Changes can be found in the revised manuscript on pages 6-8, Results section, Tables 4-7.
Comments 9: Line 82: PPF appears to be a typo.
Response 9: Thank you for pointing this out. We agree with this comment. The typo ‘PPF’ has been corrected to ‘PFF’ in the revised manuscript.
4. Response to Comments on the Quality of English Language
Point 1: (x) The English is fine and does not require any improvement.
Response 1: Thank you very much.

Reviewer 2 Report
Comments and Suggestions for Authors
Thank you for inviting me to review this paper.
My major concern is that the Introduction devotes substantial space to THA, PFF, and related complications, which—while relevant—do not directly serve the core objective of postoperative follow-up. The background on factors influencing postoperative quality of life is useful, but the reader still cannot tell whether prior follow-up studies on postoperative quality of life exist, how they were conducted, and what specific gaps remain. Although such studies clearly exist, the manuscript does not delineate their limitations; instead, it broadly states there is “a gap in understanding the quality of life of patients with PFFs,” which frames a knowledge gap about QoL rather than a targeted gap in follow-up methodology or evidence.
Second, the patient recruitment process and the choice/definition of controls are insufficiently described, leaving concerns about selection bias and the handling of age-dependent factors that affect postoperative follow-up. Although heterogeneity is noted in the limitations (page 12, line 383), these factors are fundamental to the study outcomes and should be addressed explicitly in the Methods.
Other comments:
- Page 1: The Abstract shows inconsistent line spacing between paragraphs.
- Page 2: A single-centre figure is not necessary to support an increase in PFF and may not be generalisable; consider using broader, macro-level statistics instead.
- Page 3, line 88: The abbreviation “RTH” is used before being defined.
- Page 4, line 94: “PFF” has already been defined; please avoid redefining.
- Page 4, Section 2.1: Please confirm whether patients were enrolled consecutively, specify inclusion and exclusion criteria, and provide a recruitment diagram, including the control group.
- Page 4, line 100: Provide patient demographics for each fracture type, as these are likely to influence postoperative follow-up outcomes.
- Page 4, Section 2.2: Much of the content reads as background and belongs in the Introduction; in Methods, please state the assessment tools, their frequency, and how outcomes were interpreted.
Author Response
Comments 1: My major concern is that the Introduction devotes substantial space to THA, PFF, and related complications, which—while relevant—do not directly serve the core objective of postoperative follow-up. The background on factors influencing postoperative quality of life is useful, but the reader still cannot tell whether prior follow-up studies on postoperative quality of life exist, how they were conducted, and what specific gaps remain. Although such studies clearly exist, the manuscript does not delineate their limitations; instead, it broadly states there is “a gap in understanding the quality of life of patients with PFFs,” which frames a knowledge gap about QoL rather than a targeted gap in follow-up methodology or evidence.
Response 1: Thank you for pointing this out. We agree with this comment. Therefore, we have revised the Introduction to more clearly outline the current evidence and highlight the methodological limitations of previous research. This change can be found on page 3, Introduction section, lines 71-81.
Comments 2: Second, the patient recruitment process and the choice/definition of controls are insufficiently described, leaving concerns about selection bias and the handling of age-dependent factors that affect postoperative follow-up. Although heterogeneity is noted in the limitations (page 12, line 383), these factors are fundamental to the study outcomes and should be addressed explicitly in the Methods.
Response 2: Thank you for pointing this out. We agree with this comment. Therefore, we have described patient recruitment and control selection. This change can be found on pages 4-5, Materials and Methods section, lines 118-136.
Comments 3: Page 1: The Abstract shows inconsistent line spacing between paragraphs.
Response 3: Thank you for pointing this out. We agree with this comment. Therefore, we have corrected the line spacing and have structured the Abstract according to the journal’s guidelines. This change can be found on page 1.
Comments 4: Page 2: A single-centre figure is not necessary to support an increase in PFF and may not be generalisable; consider using broader, macro-level statistics instead.
Response 4: Thank you for pointing this out. We agree with this comment. Therefore, we have removed the single-centre figure illustrating PFF cases from the Results section and incorporated the relevant information into the Materials and Methods section. This change can be found in Materials and Methods section, pages 3-4, lines 109-114.
Comments 5: Page 3, line 88: The abbreviation “RTH” is used before being defined.
Response 5: Thank you for pointing this out. We agree with this comment. Therefore, we have removed the abbreviation “RTH” from the Introduction. This change can be found on page 3, Introduction section.
Comments 6: Page 4, line 94: “PFF” has already been defined; please avoid redefining.
Response 6: Thank you for pointing this out. We agree with this comment. Redundant redefinition of ‘PFF’ has been removed throughout the manuscript.
Comments 7: Page 4, Section 2.1: Please confirm whether patients were enrolled consecutively, specify inclusion and exclusion criteria, and provide a recruitment diagram, including the control group.
Response 7: Thank you for pointing this out. We agree with this comment. Therefore, we have clarified in the Materials and Methods section that patients were enrolled consecutively, the inclusion and exclusion criteria are specified, and a recruitment flow diagram, including the control group, has been provided. This change can be found on pages 4-5, Materials and Methods section, lines 118–157.
Comments 8: Page 4, line 100: Provide patient demographics for each fracture type, as these are likely to influence postoperative follow-up outcomes.
Response 8: Thank you for pointing this out. We agree with this comment. Therefore, patient demographics for each fracture type have been added to new Table 2 (Patient baseline characteristics and follow-up) to provide a detailed overview of factors that may influence postoperative outcomes. This change can be found on page 5, Materials and Methods section, line 132.
Comments 9: Page 4, Section 2.2: Much of the content reads as background and belongs in the Introduction; in Methods, please state the assessment tools, their frequency, and how outcomes were interpreted.
Response 9: Thank you for pointing this out. We agree with this comment. Therefore, we have moved the detailed description of the instrument (SF-36) from the Materials and Methods section to the Introduction. This change can be found on page 5, Introduction section, page 3, lines 83-105.
4. Response to Comments on the Quality of English Language
(x) The English is fine and does not require any improvement.
Response 1: Thank you very much.

Reviewer 3 Report
Comments and Suggestions for Authors
The manuscript explores an important clinical topic: quality of life following Vancouver type B periprosthetic femoral fractures compared with primary THA and differences between ORIF alone and ORIF combined with stem revision. The results are clearly presented and the topic is relevant, but several methodological and interpretative aspects should be addressed to strengthen the manuscript.
-
The study includes only patients who survived and were able to complete the SF-36, which introduces both survival and selection bias. This should be explicitly acknowledged in the Methods and Limitations.
-
Please provide a brief participant flow description: total eligible fractures, exclusions (death, cognitive impairment, refusal) and final sample. A simple flow diagram would improve transparency.
-
Clarify how the control group was selected and whether their follow-up interval was comparable to the PFF group. Differences in postoperative timing may influence SF-36 scores.
-
Comorbidities, previous functional level and osteoporosis status are strongly linked to postoperative quality of life. These variables are not reported or adjusted for.
-
If available, please include baseline characteristics for both groups.
-
If multivariable analysis is not feasible, please expand the discussion on how unadjusted comparisons may limit interpretation.
-
Specify how the SF-36 was administered (self-administered, interviewer-assisted, telephone) and whether a validated translation was used.
-
Clarify how missing items were handled and how the Overall SF-36 score was calculated. There appears to be an inconsistency: overall scores in Table 5 are reported on a scale that does not match the overall scores in Table 6. Please verify and ensure consistency across the manuscript.
-
Some correlations in Table 2 (e.g., older age correlating with better general health perception and less pain) are counterintuitive. Please verify scoring direction and correct if needed. If accurate, further explanation and supporting evidence should be provided.
-
Consider reporting effect sizes and confidence intervals for key comparisons to complement p-values and improve interpretation of clinical relevance.
-
Conclusions comparing PFF patients with primary THA patients should be more cautious, given the large baseline differences between these populations.
-
For the B1 vs B2/B3 comparison, please consider whether patient selection (rather than treatment type) may partly explain the observed differences. Expand the discussion accordingly.
-
A graphical visualization of SF-36 domains (e.g., radar/spider plot) would strengthen data presentation.
-
Several sentences in the Discussion and Conclusion are long or contain repetitions; streamlining them will improve readability.
-
The manuscript is understandable but requires moderate language polishing. Several sentences are overly long, and terminology is occasionally inconsistent. A careful English revision is recommended.
-
The reference list is generally appropriate, but integration of additional recent evidence (2023–2025) specifically focused on QoL outcomes after PFF and revision arthroplasty would further reinforce the rationale and discussion.
The English in the manuscript is generally understandable, but a moderate language revision is recommended. Several sentences are overly long or contain repeated structures, reducing clarity. There are occasional issues with grammar, verb tense consistency and misuse of connectors. Terminology related to surgical procedures should be used consistently throughout the manuscript. A careful proofreading to simplify sentence structure, correct minor grammatical errors and improve flow would significantly enhance readability and presentation quality.
Author Response
Comments 1: The manuscript explores an important clinical topic: quality of life following Vancouver type B periprosthetic femoral fractures compared with primary THA and differences between ORIF alone and ORIF combined with stem revision. The results are clearly presented and the topic is relevant, but several methodological and interpretative aspects should be addressed to strengthen the manuscript. The study includes only patients who survived and were able to complete the SF-36, which introduces both survival and selection bias. This should be explicitly acknowledged in the Methods and Limitations.
Response 1: Thank you for pointing this out. We agree with this comment. Therefore, we have acknowledged in the Methods and Limitations sections that the study population was limited to patients who survived and were able to complete the SF-36 survey, which may introduce survival and selection bias. This change can be found in the revised manuscript on page 13, Discussion section (Study limitations), lines 402-410.
Comments 2: Please provide a brief participant flow description: total eligible fractures, exclusions (death, cognitive impairment, refusal) and final sample. A simple flow diagram would improve transparency.
Response 2: Thank you for pointing this out. We agree with this comment. Therefore, we have added a participant flow diagram in the Materials and Methods section, including total eligible fractures, exclusions (death, cognitive impairment), and the final sample. This change can be found in the revised manuscript on page 4, Materials and Methods section, lines 118-122.
Comments 3: Clarify how the control group was selected and whether their follow-up interval was comparable to the PFF group. Differences in postoperative timing may influence SF-36 scores.
Response 3: Thank you for pointing this out. We agree with this comment. Therefore, we have added a table summarizing baseline characteristics and follow-up information for both the PFF and THA control groups to clarify group selection and comparability of follow-up intervals. This change can be found in the revised manuscript on page 5, Materials and Methods section, lines 133-136.
Comments 4: Comorbidities, previous functional level and osteoporosis status are strongly linked to postoperative quality of life. These variables are not reported or adjusted for.
Response 4: Thank you for pointing this out. We agree with this comment. We acknowledge that comorbidities, previous functional level, and osteoporosis status are strongly linked to postoperative quality of life. Unfortunately, these variables were not reported in our dataset and could not be adjusted for in the analysis. We have now added a statement in the limitations acknowledging this potential confounding factor. This change can be found in the revised manuscript on page 13, Discussion section (Study limitations), lines 402-410.
Comments 5: If available, please include baseline characteristics for both groups.
Response 5: Thank you for this suggestion. We agree with this comment. Therefore, we have added a table presenting the baseline characteristics for both the PFF and THA control groups. This change can be found in the revised manuscript on page 4-5, Materials and Methods section, Table 2.
Comments 6: If multivariable analysis is not feasible, please expand the discussion on how unadjusted comparisons may limit interpretation.
Response 6: Thank you for this suggestion. We agree with this comment. Therefore, we have added the explanation to the Discussion section. This change can be found in the revised manuscript on page 13, Discussion section (Study limitations), lines 404-407.
Comments 7: Specify how the SF-36 was administered (self-administered, interviewer-assisted, telephone) and whether a validated translation was used.
Response 7: Thank you for pointing this out. We agree with this comment. Therefore, we have clarified in the Materials and Methods section that the SF-36 questionnaire was self-administered by patients using a validated Serbian translation. This change can be found in the revised manuscript on page 5, Materials and Methods section (Quality of life assessment and data collection), lines 139-142.
Comments 8: Clarify how missing items were handled and how the Overall SF-36 score was calculated. There appears to be an inconsistency: overall scores in Table 5 are reported on a scale that does not match the overall scores in Table 6. Please verify and ensure consistency across the manuscript.
Response 8: Thank you for pointing this out. We agree with this comment. Therefore, we have clarified in the Methods section that there were no missing items in the SF-36 questionnaires because patients completed them during routine follow-up visits in a private room, and a registered nurse assisted and verified completion if needed. The Overall SF-36 score was calculated as the mean of all eight domain scores. This change can be found in the revised manuscript on page 5, Materials and Methods section, lines 142–143.
Comments 9: Some correlations in Table 2 (e.g., older age correlating with better general health perception and less pain) are counterintuitive. Please verify scoring direction and correct if needed. If accurate, further explanation and supporting evidence should be provided.
Response 9: Thank you for pointing this out. We agree with this comment. Therefore, we have verified that the SF-36 scoring direction is correct (now Table 3). References have been added to support this and changes can be found in the revised manuscript on page 9, Discussion section, lines 232–239.
Comments 10: Consider reporting effect sizes and confidence intervals for key comparisons to complement p-values and improve interpretation of clinical relevance.
Response 10: Thank you for pointing this out. We agree with this comment. Therefore, we have calculated and included effect sizes (Cohen’s d) and 95% confidence intervals for all key comparisons to complement p-values and emphasize clinical relevance (now Tables 3–7). This change can be found in the revised manuscript on pages 6-8, Results section, Tables 3-7.
Comments 11: Conclusions comparing PFF patients with primary THA patients should be more cautious, given the large baseline differences between these populations.
Response 11: Thank you for pointing this out. We agree with this comment. Therefore, we have revised the conclusions to acknowledge the baseline differences between PFF and primary THA patients and advise caution in comparing these populations. This can be found in the revised manuscript on page 13, Conclusions section, lines 423-429.
Comments 12: For the B1 vs B2/B3 comparison, please consider whether patient selection (rather than treatment type) may partly explain the observed differences. Expand the discussion accordingly.
Response 12: Thank you for pointing this out. We agree with this comment. Therefore, we have expanded the discussion to note that differences between B1 and B2/B3 fractures may be partly influenced by patient selection rather than treatment type. This can be found in the revised manuscript on page 13, Discussion section (Study limitations) lines 406-410.
Comments 13: A graphical visualization of SF-36 domains (e.g., radar/spider plot) would strengthen data presentation.
Response 13: Thank you for the suggestion. We agree with this comment. Therefore, we have created a graphical visualization of the SF-36 domains using a radar plot to enhance data presentation. This figure can be found in the revised manuscript on page 8, Results section, Figure 3.
Comments 14: Several sentences in the Discussion and Conclusion are long or contain repetitions; streamlining them will improve readability.
Response 14: Thank you for the suggestion. We agree with this comment. Therefore, we have accordingly revised and streamlined several long or repetitive sentences in the Discussion and Conclusion sections to improve readability, with guidance from an English-language professor.
Comments 15: The manuscript is understandable but requires moderate language polishing. Several sentences are overly long, and terminology is occasionally inconsistent. A careful English revision is recommended.
Response 15: Thank you for the suggestion. We agree with this comment. Therefore, we have accordingly performed moderate language polishing throughout the manuscript, addressing overly long sentences and terminology inconsistencies.
Comments 16: The reference list is generally appropriate, but integration of additional recent evidence (2023–2025) specifically focused on QoL outcomes after PFF and revision arthroplasty would further reinforce the rationale and discussion.
Response 16: Thank you for the suggestion. We agree with this comment. Therefore, we have revised the reference list by adding several recent studies that specifically address quality of life outcomes after PFFs and revision arthroplasty. This change can be found in the revised manuscript on page 11, Discussion section, lines 318-325.
4. Response to Comments on the Quality of English Language
The English in the manuscript is generally understandable, but a moderate language revision is recommended. Several sentences are overly long or contain repeated structures, reducing clarity. There are occasional issues with grammar, verb tense consistency and misuse of connectors. Terminology related to surgical procedures should be used consistently throughout the manuscript. A careful proofreading to simplify sentence structure, correct minor grammatical errors and improve flow would significantly enhance readability and presentation quality.
Response: Thank you for pointing this out. We agree with this comment. Therefore, we have performed a moderate language revision throughout the manuscript, addressing overly long sentences, repeated structures, minor grammatical errors, verb tense inconsistencies, and inconsistent terminology. These revisions were made with the guidance of an English-language professor (Faculty of Philosophy, University of Novi Sad). This change can be found throughout the manuscript, especially in the Introduction, Methods, Results, Discussion, and Conclusion sections. All linguistic changes are highlighted in turquoise for easy identification.

Round 2
Reviewer 2 Report
Comments and Suggestions for Authors
Thank you for your efforts in revising the manuscript.
I have one remaining concern: The inclusion and exclusion criteria should be specified before the patient recruitment process—ideally within or immediately prior to Section 2.1—so that readers can clearly see how the study population was defined before viewing the recruitment flow. The patient recruitment diagram (Figure 2) should then strictly follow these criteria, rather than having a separate standalone bullet-point Section 2.3 that feels disconnected from and after the recruitment process. In addition, the recruitment process for the control groups should be described as well (e.g., source population, matching strategy, time frame), in order to minimise and transparently report potential selection bias.
Author Response
Comments 1: I have one remaining concern: The inclusion and exclusion criteria should be specified before the patient recruitment process—ideally within or immediately prior to Section 2.1—so that readers can clearly see how the study population was defined before viewing the recruitment flow. The patient recruitment diagram (Figure 2) should then strictly follow these criteria, rather than having a separate standalone bullet-point Section 2.3 that feels disconnected from and after the recruitment process. In addition, the recruitment process for the control groups should be described as well (e.g., source population, matching strategy, time frame), in order to minimise and transparently report potential selection bias.
Response 1: Thank you for pointing this out. We agree with this comment. Therefore, we have revised the manuscript - the inclusion and exclusion criteria are now specified before the patient recruitment process, within Section 2.1. The recruitment process for the control group has been described in detail. These patients were consecutively recruited from our Clinic for Orthopedic Surgery and Traumatology at the University Clinical Center of Vojvodina, with femoral neck fractures, within a predefined recruitment period. We believe these changes address your concern.
Reviewer 3 Report
Comments and Suggestions for Authors
Thank you for your thorough and thoughtful revisions. All previously raised comments have been fully and appropriately addressed. The methodological clarifications, improved data presentation and expanded discussion have strengthened the manuscript considerably. The figures and tables are clear, the statistical reporting is consistent and the limitations are well acknowledged. The manuscript is now clear, well-structured and scientifically sound.
Author Response
Comments 1: Thank you for your thorough and thoughtful revisions. All previously raised comments have been fully and appropriately addressed. The methodological clarifications, improved data presentation and expanded discussion have strengthened the manuscript considerably. The figures and tables are clear, the statistical reporting is consistent and the limitations are well acknowledged. The manuscript is now clear, well-structured and scientifically sound.
Response 1: Thank you very much for your encouraging words. It has been a great pleasure for us to have the opportunity to receive your valuable advice, which allowed us to improve both our manuscript and our research skills.